# The Potential of Small Dams for Conjunctive Water Management in Rural Municipalities

**DOI:** 10.3390/ijerph16071239

**Published:** 2019-04-08

**Authors:** Sara Soares, Daniela Terêncio, Luís Fernandes, João Machado, Fernando A.L. Pacheco

**Affiliations:** 1Department of Geology, Trás-os-Montes and Alto Douro University, Quinta de Prados, Ap. 1013, 5001-801 Vila Real, Portugal; saramsoares.03@gmail.com; 2Centro de Investigação e Tecnologias Agroambientais e Biológicas, Universidade de Trás-os-Montes e Alto Douro, Ap 1013, 5001–801 Vila Real, Portugal; dterencio@utad.pt (D.T.); lfilipe@utad.pt (L.F.); 3Vila Pouca de Aguiar Town Hall, 5450 Vila Pouca de Aguiar, Portugal; jmachado@cm-vpaguiar.pt; 4Centro de Química de Vila Real, Universidade de Trás-os-Montes e Alto Douro, Ap 1013, 5001–801 Vila Real, Portugal

**Keywords:** groundwater, surface water, small dams, conjunctive water management, hydrologic modeling, geographic information system, municipality, drinking water supply, forest, water shortage, water stress

## Abstract

The drinking water supply to Vila Pouca de Aguiar municipality in North Portugal is based on high quality groundwater, namely on nearly one hundred artesian springs and fifty boreholes. The groundwater resources are plentiful on a municipal level, but evidence some deficits at the sub-municipal (village) level, especially during the dry period (July- August) that coincides with the return of many emigrants for holiday time. The deficits affect mostly the municipal capital (Vila Pouca de Aguiar) and a neighboring village (Pedras Salgadas), which populations nearly double or even triple during that period. The estimated annual deficits approach 55,000 m^3^/yr in those villages. If the anticipated increase in consumption/habitant and decrease in annual rainfall become reality in the next two decades, then the deficits may raise to approximately 90,000 m^3^/yr. To balance the water supply system, this study proposes its transition towards a conjunctive water management based on surface water stored in small dams and groundwater. A hydrologic modeling involving small forested catchments (< 15 km^2^) elected the Cabouço watershed as most suited basin to store stream water, because surface water availability is large (2.4 Mm^3^/yr) and forest cover is dominant (84.8%). Estimated nutrient loads are also compatible with drinking water supply.

## 1. Introduction

Water demand management is a central issue in the water policy agenda [1]. The attention to this matter is related to the global population growth, increase of irrigation areas, economic development and acute water shortages in many regions around the world [2,3,4,5,6,7]. The supply of water is largely dependent on climatic factors, but management and policies play an immense role on the demand side of water systems [8,9]. Climate change and socioeconomic factors have increased the complexity of urban water supply systems [10,11,12]. Thus, fresh water sources are being gradually diversified [13], and even include recycled water in some regions [14]. Conjunctive water management involves the combined use of groundwater, surface water and/or additional sources of water to achieve public policy and management goals. Conjunctive water management enables greater water supply security and stability, helps adaptation to climate variation and uncertainty and reduces depletion and degradation of water resources [15].

The drinking water supply to Vila Pouca de Aguiar, a municipality located in the north of Portugal and composed of small villages, is currently based on groundwater. In general, the villages are supplied with perennial spring water complemented with groundwater pumped from boreholes during the dry summer months (June-September). The months of July and August concur with the return of many emigrants for holidays, which doubles or even triples the population in some villages. To account for the concomitant water demand, the artesian wells need to be more intensely pumped, because springs in this period are undergoing the recessive period with progressively lower discharge rates. In the long term, the excessive pumping at the drilled wells is expected to cause borehole exhaustion and aquifer over-exploitation. The steady reduction of precipitation over the past century is likely to add negative effects on groundwater resources, namely on annual spring yields and borehole productivity. The combined effects of well-exhaustion and rainfall reduction are leading the municipal system of public water supply into a progressively more fragile situation.

A drinking water supply system based on groundwater is inestimable because the quality of groundwater resources is usually very high. The system used in Vila Pouca de Aguiar, mostly based on granite bearing shallow groundwater [16,17], is no exception. However, the aforementioned periodical shortage problem needs to be solved. Eventually, the solution relies on moving the system to the conjunctive water management era and in that context on complementing the groundwater supply with an alternative source. In this study, surface water diverted into small dams in specific forested basins is the proposed complementary source.

The storage of stream water in small reservoirs is likely to be an appropriate complement to drinking water supply systems based on groundwater, such as the Vila Pouca de Aguiar system. Catchments, even if they are relatively small, are apt to provide substantial volumes of stream water. Eventually, these volumes can compensate the spring deficits observed in summer, and concomitantly protect drilled wells from exhaustion and aquifers from over-exploitation. However, dam locations need to comply with a number of physical, ecological and socioeconomic pre-requisites, as detailed in the recent papers of Terêncio’s group [18,19]. A special attention is ought to dominant land uses and corresponding natural uses to avoid the water quality impacts of land use conflicts [20,21], as well as to biophysical processes controlling stream water quality [22,23].

Despite the large number of papers addressing conjunctive water management and the recent development of spatially distributed models to allocate stream water collection systems in catchments, the combined assessment of both topics in a single study is less frequent. The purpose of this study is therefore to present an assessment of groundwater availability and public water consumption in the municipality of Vila Pouca de Aguiar, coupling the results with a suitability analysis of potential small dam sites, aiming the evaluation of annual groundwater deficits and the identification of areas that require a drinking water complement with surface resources.

## 2. Materials and Methods

### 2.1. Study Area

The region of Vila Pouca de Aguiar is located in the north of Portugal and occupies an area of approximately 437 km^2^ (Figure 1). The morphology is characterized by large-scale tectonic valleys associated with the Vila Real fault. These valleys are surrounded by the Alvão (to the West) and Padrela (to the East) mountains. Altitudes range from a minimum of 320 m in the northern valleys to a maximum of 1130 m up in the mountains. The geology is characterized by Hercynian (syn- to post-tectonic) granites that intruded Palaeozoic (Cambrian to Devonian) metasediments and were covered by Quaternary alluvial and terrace deposits along the Vila Real fault. In the Southeast part of the area, occupied by an extensive outcrop of metasediments, a geological structure was defined, in which tectonic laminae are folded and separated by major thrusts [16,17].

Climate in the area is temperate with alternating wet–cold (October–April) and dry–warm (May–September) seasons. The long-term average precipitation ranges from 900 mm/yr in the northeast to 1900 mm/yr in the southwest of the region, being influenced by the topography. In the past seven decades precipitation has steadily decreased at a fairly constant average rate of 6.9 % per decade (Figure 2). Groundwater composition is dominated by water-mineral interactions in granite and metasediment bedrock [24,25,26,27,28,29]. Despite the potential threats from anthropogenic sources, such as agriculture or domestic effluents, the chemical fingerprint of groundwater is natural [30,31] and dominated by sodium and bicarbonate. On average, concentrations of dissolved compounds usually linked to anthropogenic activity, such as chloride, nitrate or sulphate, are low: [Cl^–^] = 4.4 mg/L, [SO_4_^2–^] = 1.0 mg/L and [NO_3_^–^] = 0.7 mg/L [17].

In the municipality of Vila Pouca de Aguiar, the public supply of water is accomplished with groundwater resources, namely artesian springs, drilled wells and an alluvial well (Figure 3). This water supply infrastructure has been built for over 30 years and represents an asset of great economic, environmental and social value to the community.

The annual discharge of approximately one hundred springs reaches 1.25 Mm^3^, but spring discharges vary considerably within the municipality (Figure 4a). In the NW and NE sectors the annual discharge is frequently < 5000 m^3^ per spring while in the SE sector it can reach 100,000 m^3^. The variation of spring discharge is also expressed in time. The lowest discharges occur in August (Figure 4b) and vary from 300 to 2900 m^3^/month. In this month the municipal capacity, considering the contribution of all springs, is close to 83,000 m^3^. This value is 20% lower than the average monthly discharge (≈ 104,000 m^3^/month).

The productivity of drilled wells is illustrated in Figure 4c, being influenced by local precipitation. In the NE sector where precipitation is < 900 mm/yr the drilled well productivity is < 2 m^3^/h. As one moves towards the SW sector and precipitation increases up to 1,900 mm/yr, the drilled well productivity also increases reaching 10–20 m^3^/h in various areas. The alluvial well was installed in the Torno River that ensures 0.5 Mm^3^/yr of groundwater supply. This well, with a 2.5 m diameter, was dug in the riverbed until 7–8 m depth. The digging crossed the alluvial deposit and the underlying granite. In the alluvial deposit, a couple of 5–6 m long and 40 cm wide horizontal drains were installed along the river’s upstream and downstream directions and linked to the well. These drains were made of fiberglass covered by geo textile and were installed over a gravel bed. At the bottom of the well, a couple of 50 m long horizontal wells were drilled in the granitic bedrock to enhance groundwater diversion towards the well. In both cases, the drilling was executed with 7.5 cm diameter.

The consumption of groundwater in Vila Pouca de Aguiar has almost doubled in the past two decades (Figure 5a), from approximately 0.6 Mm^3^/yr in 1995 to 1.1 Mm^3^/yr in 2011 (https://www.pordata.pt/). The average percent increase was 4.4%/yr. The growth is not related to population increase, on the contrary, the population has decreased by some 2000 people during the same period (from 15,000 to 13,000 inhabitants, approximately). The cause of consumption increase, from 29 m^3^/yr/habitant to 51 m^3^/yr/habitant, was therefore the expansion of comfort water uses, such as mechanical dishwashing, irrigation of private gardens, car washing, among others. As illustrated in Figure 5b, total consumption comprises the water legally discharged from the public network and paid by the residents (61%), the undetected leakages or clandestine consumption (10%) and the public consumption related to irrigation of municipal gardens, maintenance of municipal swimming pools, water supply to public interest organizations (local sports clubs, firefighter corps), among other uses (29%). In the context of conjunctive water management the groundwater currently used for garden irrigation should be replaced by harvested rainwater or treated wastewater [32,33,34,35,36].

The consumption of groundwater is substantially different in the months of July and August, when compared to the other calendar months, because July–August is the period when most emigrants return for holiday time. During this stage the population almost doubles in the municipality and can even triple in the capital (Vila Pouca de Aguiar town). The consumption of groundwater increases proportionally. Therefore, in July and August the water consumption is estimated in 157,000 m^3^/month while in the other months is half of that value.

The spatial distribution of total consumption is portrayed in Figure 6. Approximately 40% of all groundwater consumed in the municipality occurs in the capital (Vila Pouca de Aguiar) and Bornes de Aguiar/Pedras Salgadas town, which is a consequence of population concentration in the larger villages. A thorough inspection of this figure reveals a striking reality. The water consumption in villages from the Jales plateau (Vreia de Jales and Alfarela de Jales) are included in the lowest category (< 30,000 m^3^/yr), while this region is located where springs can naturally discharge over 100,000 m^3^/yr of shallow groundwater. In opposition, consumption in Vila Pouca de Aguiar (≈ 257,000 m^3^/yr), Bornes de Aguiar/Pedras Salgadas (≈ 200,000 m^3^/yr), Capeludos (≈ 89,000 m^3^/yr) and Telões (≈ 85,000 m^3^/yr), are included in the highest categories, while these villages are located where average spring discharge is low (5000–20000 m^3^/yr).

### 2.2. Conceptual Approach

The relationship between groundwater resource availability and consumption in the municipality of Vila Pouca de Aguiar is summarized in Table 1. At a municipality level, the resource availability far exceeds the water demand. The overall water surplus is > 1 Mm^3^/yr and occurs during the wet season. The excess water from springs is discharged through stormwater discharge tubes located in the upper part of drinking water tanks and naturally diverted to nearby streams. The excess water from Torno River reservoir is released through overflow devices installed in the weir. Finally, the 8 h of pumped borehole water cannot be used during the wet season.

The groundwater surplus scenario is not reproduced in the larger villages, especially in the dry season. The smaller springs and scenario of Torno River groundwater discharges coupled with the temporary population increase (return of emigrants) in the months of July and August have caused water deficits in Vila Pouca de Aguiar and Bornes de Aguiar/Pedras Salgadas towns. In 2011, these defices were > 55,000 m^3^ (Table 1). In general, these deficits have been satisfied by the municipal authorities through translocation of water from the Alvão region using vehicle water tanks. The use of water from neighbouring large dams (e.g., the Pinhão River dam with a storage capacity of 4.2 Mm^3^; http://www.somague.pt/portfolio_detail/barragem-do-pinhao/) has not been attempted. Normally, the quality of surface water in large dams is poor because of multiple anthropogenic inputs [37,38] and therefore requires expensive treatment [39]. Besides the quality issue, the large dam solution would eventually imply a loss of municipal autonomy concerning drinking water supply to the population, because large dams are frequently exploited by private companies. It is therefore urgent to set up alternative strategies to balance water resource availability and demand in every village or town within the Vila Pouca de Aguiar municipality. This exigency should be taken more seriously if projected water deficits > 85,000 m^3^/yr become true, related to expected 5% consumption increase and 15% precipitation decrease over the next two decades (Table 1).

In this study, the proposed alternative source was stream water stored in small dams installed in small catchments and explored by the municipal authorities. The selection of sites was based on three suitability parameters: water availability, dominant land cover and altitude difference between the catchment outlet and drinking water tanks used to supply the towns of Vila Pouca de Aguiar and Bornes de Aguiar/Pedras Salgadas. The suited sites will be located where high rainfall ensures large stream flows, forest land cover makes certain good quality water, and the altitude of catchment outlets allows the transport of water by gravity to the application areas reducing costs. The selection of suitable catchments relied on hydrologic modeling (ArcSWAT) for determining water availability and Geographic Information System (GIS) assessments for evaluating the aforementioned land cover and altitude difference parameters, as detailed in the next sections.

### 2.3. Hydrologic Model and GIS Assessments

The availability of surface water in catchments and groundwater in underlying aquifers can be accessed from a diversity of models [40,41,42,43,44,45,46]. The present study resorted to the SWAT model [47], which simulates river flows based on meteorological data (rainfall, temperature, wind speed, solar radiation, relative humidity) and information on the catchment’s physical characteristics (topography, hydrologic network, soils) and vegetative cover/use. The ArcSWAT software (https://swat.tamu.edu/software/ arcswat/) implements the SWAT model in GIS (Figure 7). Firstly, the ArcSWAT delineates the catchment’s hydrologic network from the reading and interpretation of a digital terrain model. At this stage, the hydrographic density is selected by the user. Having completed this operation the software draws a set of water lines and their intersections, also drawing the basins outlets and boundaries. Subsequently, the ArcSWAT links the watercourses located upstream from the basin mouth to their sub-basins and then subdivides the sub-basins into the so-called hydrological response units (HRUs). The HRUs are homogeneous relative to topography (slope), soil and land cover. The sub-basins and HRUs are then characterized for geometric properties (length, width, area, perimeter, mean slope), hydrologic response (curve number), and meteorological settings (usually, daily precipitation records and average values for the other relevant parameters). At the end of this geometric, hydrologic and meteorological characterization, the flow components associated to each HRU are calculated using water balance equations, flow models and routing algorithms adapted to the surface, sub-surface and underground flow conditions. Based on the indicated land uses, nutrient loads (e.g., nitrate) are also estimated using solute transport models. This step includes the estimation of total river flows and nutrient concentrations at the basin outlet, which are iteratively adjusted to measured counterparts through a calibration procedure. Finally, the calculated flow components (e.g., runoff, groundwater) and nutrients are assigned to the respective sub-basins and associated outlets.

The GIS assessments comprised the digital sampling of forest cover percentages in each modeled catchment (using the “*Tabulate Area*” tool of “Spatial Analyst > Zonal” toolbox of ArcGIS), and the digital sampling of drinking water tank altitudes located in the Vila Pouca de Aguiar and Bornes de Aguiar/Pedras Salgadas towns, as well as of altitudes at the catchment outlets (using the “*Sample*” tool of the same toolbox). Having sampled the aforementioned altitudes, the differences between the catchment altitudes and the average altitude at the drinking water tanks were calculated with the ArcGIS “*Field Calculator*” tool.

### 2.4. Data and Software

The Digital Terrain Model used to characterize the relief and delineate the catchments, sub-catchments, hydrologic networks and outlets, is a 25 × 25 m resolution raster map that was downloaded from the of the Portuguese Geographic Institute (http://www.igeo.pt). This geographic data is referenced in the Datum 73 system and Cascais Altimetric Datum, Hayford Ellipsoid. The soil characterization was based on cartography and related datasheets published in digital format by the Emergency Information System Network website (http://scrif.igeo.pt). This information is referenced in the Datum International 1924 system, Transverse Mercator projection. The CORINE Land Cover map for continental Portugal (CLC 2006, http://www.dgterritorio.pt/dados_abertos/clc/) was used to characterize vegetation cover. This digital map is referenced in the Datum Lisbon system, Hayford ellipsoid, Transverse Mercator projection. The meteorological data (daily precipitation and long-term average temperature, humidity, wind speed and solar radiation) were downloaded from the Portuguese Water Resources Information System (http://snirh.pt/), and span the 2003–2007 period. The coordinates of meteorological stations are referenced in the Datum Lisboa system, Hayford ellipsoid, Transverse Mercator projection. All the digital data are based on rectangular coordinates with Gaussian projection. The softwares used to handle all the geographic data were ArcGIS (http://www.esriportugal.pt/) and ArcSWAT (https://swat.tamu.edu/software/arcswat/), two common tools used in hydrologic and environmental studies [48,49,50,51].

## 3. Results

The geographic data used to run the ArcSWAT simulations are illustrated in Figure 8. Figure 8a shows the modeled catchments (all perennial streams with area < 15 km^2^), the associated hydrologic networks and basin outlets. It also represents the meteorological stations. In the tested period (2003–2007), precipitation reached 1036 ± 653.9 mm/yr, while evaporation losses were 715.9 ± 169.4 mm/yr. Figure 8b describes the spatial distribution of major soil types, which correspond to luvisols and cambisols. Figure 8c depicts land cover represented by major types (e.g., agriculture, forest). Figure 8d illustrates the spatial distribution of terrain slope.

The modeling results are summarized in Figure 9. Figure 9a describes surface water while Figure 9b describes groundwater. In either case the annual volumes represent non calibrated values, because no hydrometric stations were available to use in the calibration step. In the modeled catchments, the maximum available surface water can reach 5 Mm^3^/yr (Ribeiro do Boco), while the maximum storable groundwater will not exceed 3 Mm^3^/yr (Ribeiro do Boco and Ribeiro do Cabouço). These values do not account for potential evaporation losses directly from the reservoir, following dam construction. They also neglect losses related with ecological flows. Regardless the catchment, the storage of surface water or groundwater would fulfill present day water deficits and future demands, even considering the aforementioned losses. The percentage of forest cover in the modeled catchments is represented in Figure 10. The percentages span a wide range of values (from 49.1 to 88.8%).

The range of altitude differences between the catchment outlets and the drinking water tanks located in the deficit villages of Vila Pouca de Aguiar e and Bornes de Aguiar/Pedras Salgadas are also quite ample, which points to substantial differences between the catchments considering this suitability parameter.

A combined assessment of catchment suitability based on the volumes of surface water discharge, percentage of forest cover and altitude difference between catchment outlet and drinking water tanks is tempted in Table 2. In this suitability exercise, catchments were ranked according to surface water availability (rank 1), percentage of forest cover (rank 2) and altitude difference (rank 3), and then the three ranks were added (global rank). According to this combined assessment, the most suitable catchment to install a small dam aiming the complement of drinking water supply to the villages of Vila Pouca de Aguiar and Bornes de Aguiar/Pedras Salgadas is the Cabouço catchment. This basin was ranked 7 on surface water availability (2.4 Mm^3^/yr), 8 on forest cover (84.8%) and 6 in altitude difference (86.8 m), and would be a nice place to store good quality water in the Vila Pouca Municipality. The use of other catchments is questionable given their low positions in one or more ranks. For example, the Ribeiro do Carvalhal and Ribeiro do Boco catchments are highly positioned in Rank 1 – water availability (positions 8 and 9, respectively), but the negative altitude differences reduces their positions in Rank 3 (positions 1 and 2, respectively) virtually impeding their selection for dam installation considering the potentially high water transport costs. Other catchments are also affected by negative altitude differences, such as the Ribeiro da Peliteira and Ribeiro de Revel (positions 4 and 3 in Rank 3). In these cases, dam installation suitability is further hampered by the low positions in Rank 1 (5 and 2). Rank 2 may hinder the use of Rio Pinhão and Ribeiro dos Rebujais catchments because of their limited forest cover (≈ 50%; positions 2 and 1). A similar rationale holds for Rio Tinhela catchment (64.6% of forest cover, position 5 in Rank 2). Finally, the Ribeiro do Torno catchment is positioned high in ranks 2 and 3 but low in Rank 1.

## 4. Discussion

The historical drinking water supply data of the municipality of Vila Pouca de Aguiar reveals a generally sustainable system because at a municipal level the explored groundwater resources exceed the peoples’ and public consumption. However, some local deficits were detected in two dry months (July-August) and in larger villages (Vila Pouca de Aguiar and Bornes de Aguiar/Pedras Salgadas), amplified by the return of emigrants during this period. The actual deficits are greater than 55,000 m^3^/yr and could increase to approximately 90,000 m^3^/yr in the near future (next two decades) if the rainfall decrease (–7.5%/decade) and consumption increase (+2.5%/decade) scenarios become reality, as can be anticipated from past trends (Figure 2 and 5a, respectively). The results of a study conducted in Denmark also highlighted that freshwater impact assessments based on regional data, rather than local data, may dramatically underestimate the actual impact on the water resource [52]. Some other studies generally discussed the role of scale on water scarcity assessments [53,54,55].

The general surplus of groundwater resources at the municipal scale could raise the hypothesis of groundwater diversion from regions of surplus to regions of deficit. This has been tempted in Vila Pouca de Aguiar, but with limited success. Local people and authorities usually look at the artesian springs and boreholes explored in their territories as their property, and barely consider the possibility to share this water with neighbor deficit areas. The political debate between parties tends to exacerbate this issue. To our view, the option is relevant but requires modification of governance in directions comparable to other cases [56,57,58,59].

The alternative to balance water resource availability and demand in the Vila Pouca de Aguiar municipality, proposed in this study, is to complement the installed groundwater system with an adequate surface water system, following the conjunctive water management rationale and guidelines [15]. This surface system is to be installed in areas where rainfall is high and land cover is predominantly forest ensuring good quality. At these sites, stream is stored in small natural reservoirs created by small dams and diverted to the deficit areas. Water diversion is a common practice at river scale [60]. In the Vila Pouca de Aguiar municipality, the Cabouço catchment was considered the most suited basin to install a small dam, because surface water availability was large (2.4 Mm^3^/yr) and forest cover was dominant (84.8%). Given the location of Cabouço’s outlet at 900 m altitude, the water transfer into the drinking water tanks from the Vila Pouca de Aguiar and Bornes de Aguiar/Pedras Salgadas villages, located below 870 m of altitude, will be gravitational reducing the diversion costs.

The Cabouço catchment can deliver 2.4 Mm^3^/yr of surface water to the outlet but it also discharges 1.5 Mm^3^/yr of groundwater to the same point (Table 2). Therefore, the availability of clean water resources in the catchment is 3.9 Mm^3^/yr. Actual and forecasted deficits are < 0.1 Mm^3^/yr (Table 1), and hence some water could be released for other uses. The municipality of Vila Pouca de Aguiar comprises a 12.1 km^2^ valley used for agriculture, termed the “South Valley” (the green shaded area in Figure 10). The water for irrigation is usually pumped from the Corgo River that crosses the valley, or from drilled wells. Farmland area and productivity in this valley would eventually increase if crops could be irrigated with water from the Cabouço catchment, since the location is highly favorable (Figure 10).

The creation of the Cabouço dam reservoir would also contribute to local aquifer recharge, and hence improve productivity of boreholes downwards. Other studies refer to stream damming as a measure to develop aquifer recharge [61,62]. The quality of stream water is another important issue, because its purpose is to use it for drinking water. Water quality has not been measured in the Cabouço catchment in this study. ArcSWAT provided estimates for annual nitrate loadings (Kg N/ha) in surface water and groundwater (Figure 9) based on catchment land cover, which can be used as indicators of water quality. The estimated loads are higher in groundwater (4.7 Kg.N/ha) than in surface water (0.4 Kg.N/ha), but are generally low. It is very important to keep nutrient loads low in dam reservoirs to prevent water quality deterioration and associated impacts, namely eutrophication, biodiversity decline and general ecosystem degradation [63,64,65]. It is worth to note that the estimated nitrate loads are rough indications of real loads because they could not be calibrated with measured data, and therefore a specific dedicated study should be performed for a better assessment.

Overall, the conjunctive management of Cabouço stream water and the installed system based on groundwater seems reliable for Vila Pouca de Aguiar in the long-term. It is also strategically appropriate because it maintains municipal autonomy over the public drinking water supply. In this context, it is worth mentioning the need to develop public policies and implement land management plans in a manner that Cabouço area can be preserved from water pollution and legally protected for conservation of water resources. This study exposed the interdependency, availability, and accessibility of surface water and groundwater in Vila Pouca de Aguiar, and therefore contributed to identify and manage water security in the municipality [66]. We are therefore confident that the construction of Cabouço dam to complement drinking water supply in deficit areas and complementary irrigate the South Valley, would represent the most reliable route to follow in achieving sustainable use of local water resources.

## 5. Conclusions

In this study, the proposal of conjunctive water management for drinking water supply of a small rural municipality proved efficient. The studied area comprised the Vila Pouca de Aguiar municipality located in the north of Portugal, which uses high quality groundwater as source for drinking water supply. Although the groundwater resources are abundant at municipal level, some deficits were detected at sub-municipal levels, in July and August, reaching > 55,000 m^3^/yr in the two largest villages (the municipal capital Vila Pouca de Aguiar and the Pedras Salgadas town). These deficits are recurrent and therefore this study investigated the possibility to reinforce the system with stream water stored in small dams. The hydrologic model (ArcSWAT) of small forested catchments (< 15 km^2^) located within the municipality limit revealed the suitability to store stream water in the Cabouço catchment. This 7 km^2^ catchment can deliver approximately 3.9 Mm^3^/yr of good quality water, because of its strategic location in a densely forested area (> 85% forest and shrub cover). The location at an altitude of 900 m enables the gravitational transport of water towards the application areas (mostly Vila Pouca de Aguiar and Pedras Salgadas towns), at low diversion costs. Overall, this solution would bring the public supply of drinking water in Vila Pouca de Aguiar into the conjunctive water management era, ensuring the system’s sustainability for the future generations.

## Figures and Tables

**Figure 1 ijerph-16-01239-f001:**
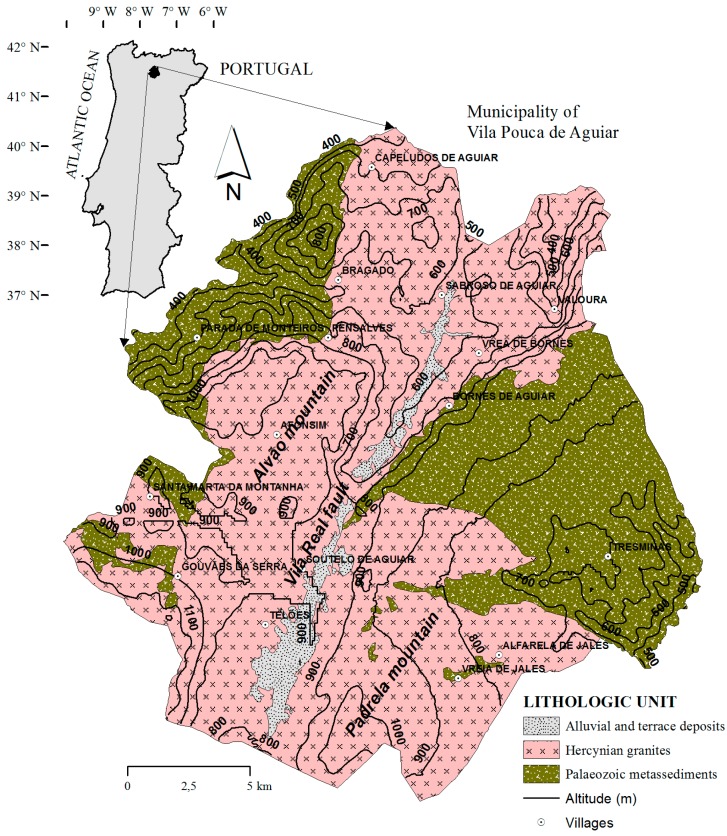
Location, geology and altitude contours in the Vila Pouca de Aguiar municipality.

**Figure 2 ijerph-16-01239-f002:**
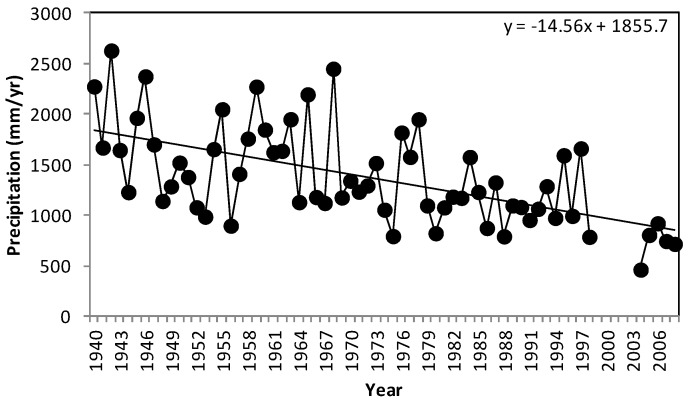
Evolution of annual precipitation in the past seven decades in the Vila Pouca de Aguiar municipality. The precipitation data were compiled from the Portuguese Water Institute (https://snirh.apambiente.pt) and refer to National station 05L/01UG.

**Figure 3 ijerph-16-01239-f003:**
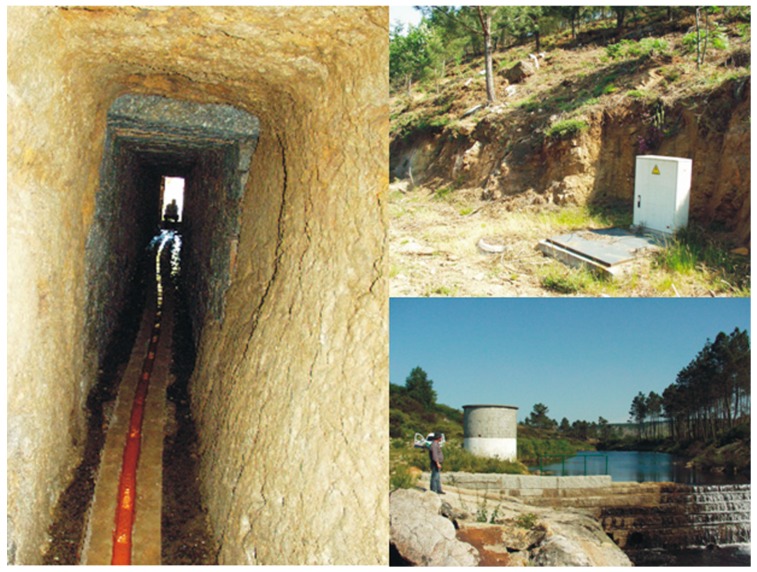
Sources of groundwater used in Vila Pouca de Aguiar for public supply of drinking water: left panel—spring/gallery; upper right panel—drilled well; lower right panel—alluvial well.

**Figure 4 ijerph-16-01239-f004:**
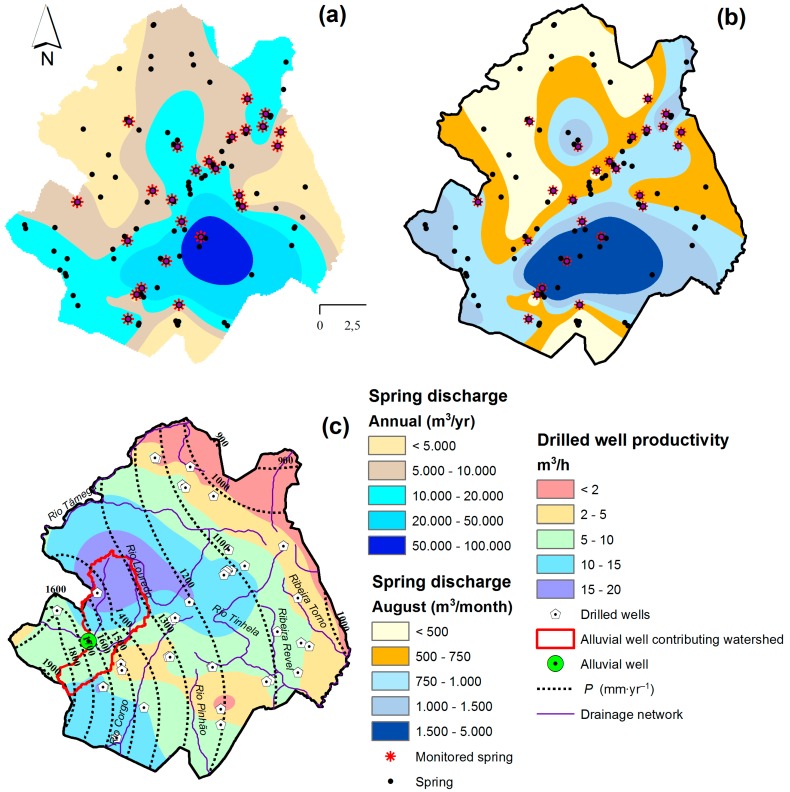
(**a**) Spatial distribution annual spring discharge. The shaded areas were drawn (interpolated) on the basis of discharges measured at the monitored springs. The discharge at the other springs (black circles) was estimated through digital sampling of interpolated discharges at the spring site. (**b**) Spatial distribution of August’s spring discharge. This is the month when discharges are lowest. (**c**) Spatial distribution of drilled well productivity, with indication of drilled and alluvial well locations, the alluvial well contributing area, the drainage network and the contours of annual precipitation. The source data were compiled from [16,17].

**Figure 5 ijerph-16-01239-f005:**
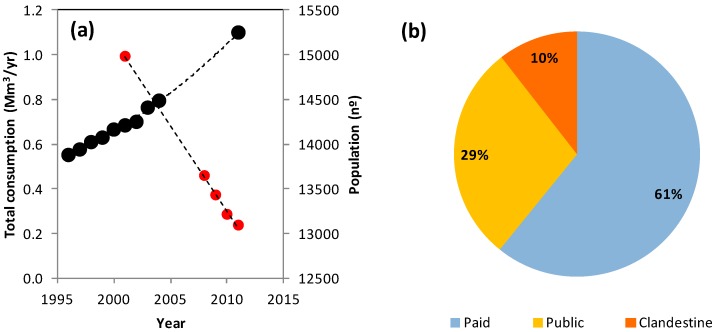
(**a**) Consumption of groundwater in Vila Pouca de Aguiar in the period 1995–2011, based on municipal records. Evolution of resident population in the period 2001–2011 (https://www.pordata.pt/). (**b**) Groundwater consumption as function of type: paid by residents, public use (unpaid) and clandestine or related to network losses.

**Figure 6 ijerph-16-01239-f006:**
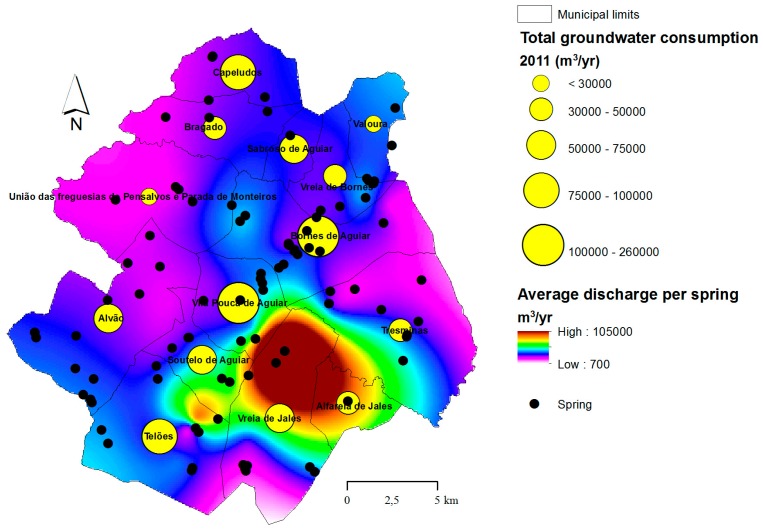
Spatial distribution of groundwater consumption within the Vila Pouca de Aguiar municipality (graduated yellow circles). The circles are plotted over the spatial distribution of annual spring discharge (average yield per spring) to expose the distortion between resource availability and demand. For example, Vila Pouca de Aguiar has the largest demand for groundwater but is located where spring discharge is low. Conversely, Alfarela de Jales or Vreia de Jales are low demanding villages located in a region with large resource availability.

**Figure 7 ijerph-16-01239-f007:**
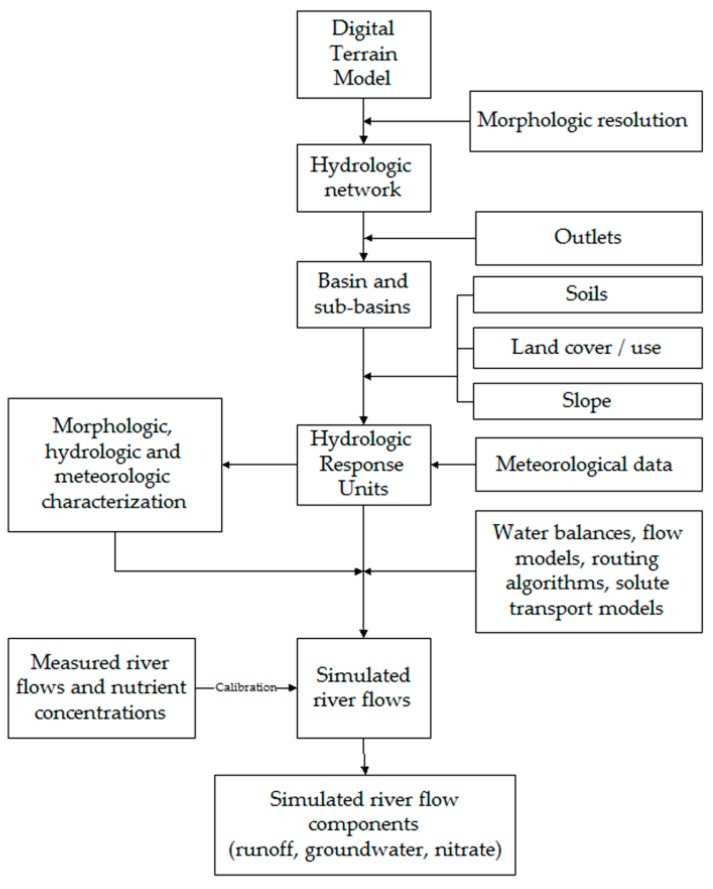
General flowchart of ArcSWAT.

**Figure 8 ijerph-16-01239-f008:**
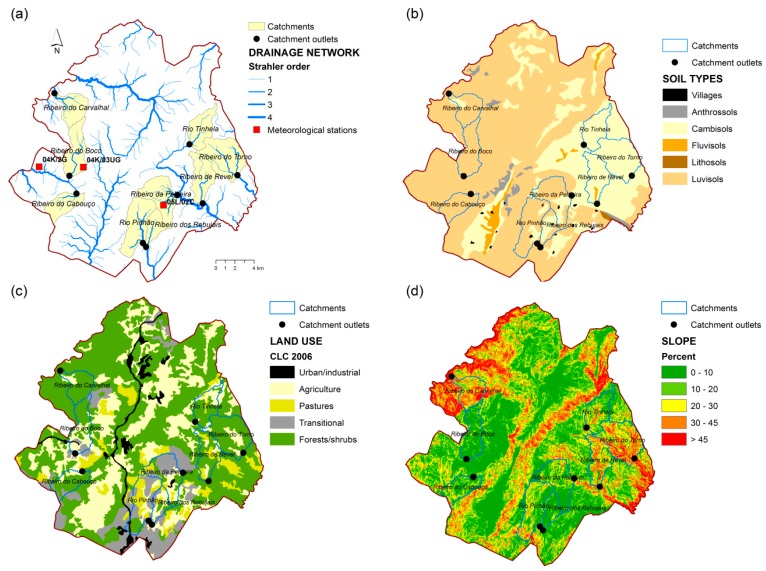
Geographic data used in the hydrologic simulations based on ArcSWAT. (**a**) Hydrographic and meteorological data; (**b**) Soil data; (**c**) Land use data; (**d**) Topographic data.

**Figure 9 ijerph-16-01239-f009:**
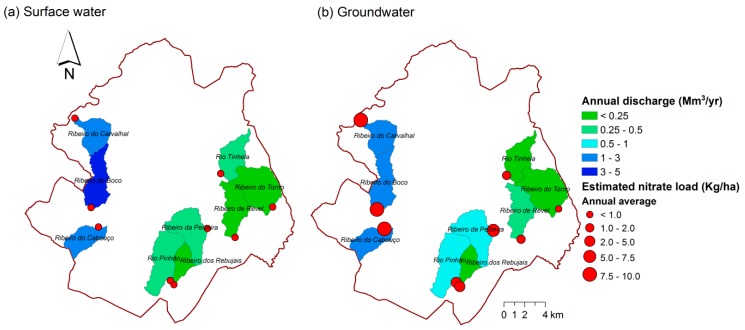
ArcSWAT results. (**a**) Volume of naturally discharged surface water and of corresponding nitrate load; (**b**) volume of naturally discharged groundwater and corresponding nitrate load.

**Figure 10 ijerph-16-01239-f010:**
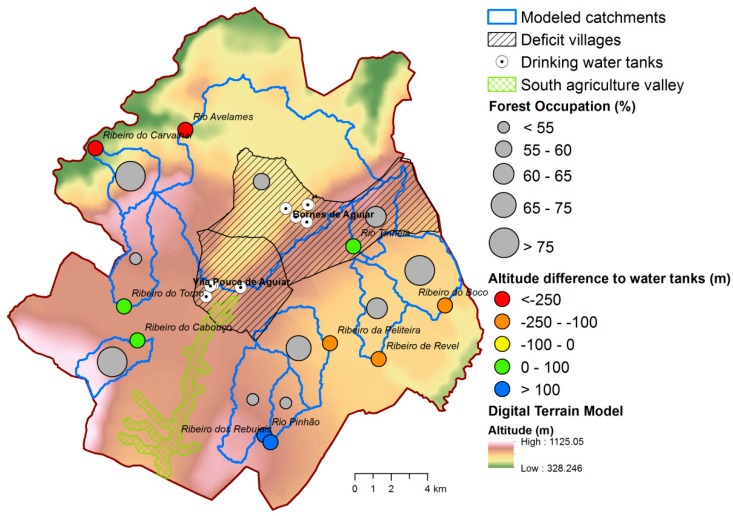
Percentage of forest cover (graduated grey circles) in the modeled catchments represented by the blue boundaries. Altitude differences between catchment outlets and the drinking water tanks (colored circles) located in the deficit villages of Vila Pouca de Aguiar and Bornes de Aguiar/Pedras Salgadas represented by the black shaded areas. The South Valley used extensively for agriculture is represented by the green shaded area. All this information is plotted over the municipal digital terrain model.

**Table 1 ijerph-16-01239-t001:** Relationship between groundwater resource availability and groundwater consumption in the entire Vila Pouca de Aguiar municipality and the two largest towns, considering the annual resources and total consumptions as well as the July–August counterparts. The scenario “5–15” refers to the expected 5% consumption increase and 15% precipitation decrease over the next two decades.

Source	Resource Availability (m^3^)			
	Municipality		Vila Pouca de Aguiar and Bornes de Aguiar/Pedras Salgadas	
	Annual	In July and August	Annual	In July and August
Springs	1,246,123	166,188	317,580	21,017
Drilled wells (8 h pumping every day)	831,032	141,162	188,340	31,992
Alluvial well	470,844	62,779	470,844	62,779
	Resource demand (m^3^)			
Total consumption in 2011	1,099,302	314,086	457 332	171,500
	Balance (Surplus/deficit)			
Actual situation	1,448,697	56,043	519,432	–55,711
Scenario “5–15”	1,156,402	–9577	401,994	–87,226

**Table 2 ijerph-16-01239-t002:** Combined assessment of catchment suitability based on availability of surface water (Rank 1), percentage of forest cover (Rank 2) and altitude difference between catchment outlets and drinking water tanks located in the Vila Pouca de Aguiar and Bornes de Aguiar/Pedras Salgadas villages.

Identification and Characterization			Water Availability			Location Constraints				Global Rank
Catchment	Area (km^2^)	Long-Term Precipitation (mm/yr)	Surface Water (Mm^3^/yr)	Groundwater (Mm^3^/yr)	Rank 1	Forest Cover (%)	Rank 2	Altitude Difference (m)	Rank 3	
Ribeiro do Cabouço	7.1	1405.9	2.4	1.5	7.0	84.8	8.0	86.8	6.0	21.0
Ribeiro do Carvalhal	7.9	1405.9	2.7	1.3	8.0	88.8	9.0	-411.2	1.0	18.0
Rio Pinhão	13.0	815.4	0.4	1.0	6.0	52.1	2.0	138.2	9.0	17.0
Ribeiro do Torno	14.6	665.2	0.2	0.2	3.0	81.3	7.0	87.0	7.0	17.0
Ribeiro da Peliteira	11.8	815.4	0.4	0.9	5.0	66.3	6.0	-108.7	4.0	15.0
Rio Tinhela	8.3	815.4	0.3	0.2	4.0	64.6	5.0	73.4	5.0	14.0
Ribeiro do Boco	9.8	1405.9	3.4	2.1	9.0	53.0	3.0	-190.7	2.0	14.0
Ribeiro dos Rebujais	4.8	815.4	0.1	0.2	1.0	49.1	1.0	126.3	8.0	10.0
Ribeiro de Revel	9.3	815.4	0.2	0.3	2.0	62.9	4.0	-115.8	3.0	9.0

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
