# Peer review of "The Potential of Small Dams for Conjunctive Water Management in Rural Municipalities"

_ijerph, 2019, doi:10.3390/ijerph16071239_

Round 1

Reviewer 1 Report

Summary

The study investigates the use of rainwater harvesting in an area north of Portugal.

Broad comments

The term rainwater harvesting in this manuscript seems to be quite misleading as the authors proposes water to be diverted from a river and stored in a dam. Rainwater harvesting is often described to be rainwater captured with minimal runoff and stored locally. The catchment options in the manuscripts with sizes of >5km2 are not considered to be localised capture of rainwater. Please consider changing the title and the associated wordings within the manuscript.

Please check the English in the manuscript as there are a number of grammatical and spelling errors. For instance, the use of “duplicate” and “triplicate” has been used through the manuscript. This is incorrect use of the words. Please change to “double” and “triple”, respectively.

Specific comments

Line 135 – 29 m3/yr.habitant should be 29 m3/yr/habitant. Please change accordingly throughout manuscript.

Line 271 – A brief summary of the parameters that went into the hydrological model ArcSWAT should be provided, including rainfall data period of analysis, and loss parameters to support the volume produced through surface runoff.

Line 330 – As the water is stored in dams, has the study accounted for evapotranspiration losses?

Line 337 – Please provide reasons as to why the other dams may not be as suitable as the chosen option.

Line 352 – Should it be the nitrates loads are higher in surface water (4.7 kg.N/ha) than in groundwater (0.4 Kg.N/ha)?

Author Response

Reviewer #1

Summary

The study investigates the use of rainwater harvesting in an area north of Portugal.

Broad comments

Comment #1

The term rainwater harvesting in this manuscript seems to be quite misleading as the authors proposes water to be diverted from a river and stored in a dam. Rainwater harvesting is often described to be rainwater captured with minimal runoff and stored locally. The catchment options in the manuscripts with sizes of > 5km2 are not considered to be localised capture of rainwater. Please consider changing the title and the associated wordings within the manuscript.

Authors response

We thank the reviewer comment. We changed the title and adapted the entire manuscript according to the reviewer comment. The title is now

The potential of small dams for conjunctive water management in rural municipalities

The changes to the manuscript are highlighted in yellow.

Comment #2

Please check the English in the manuscript as there are a number of grammatical and spelling errors. For instance, the use of “duplicate” and “triplicate” has been used through the manuscript. This is incorrect use of the words. Please change to “double” and “triple”, respectively.

Authors response

The aforementioned corrections were done. The entire manuscript was thoroughly checked to detect and correct other grammatical and spelling errors.

Specific comments

Line 135 – 29 m3/yr.habitant should be 29 m3/yr/habitant. Please change accordingly throughout manuscript.

Authors response: Done

Line 271 – A brief summary of the parameters that went into the hydrological model ArcSWAT should be provided, including rainfall data period of analysis, and loss parameters to support the volume produced through surface runoff.

Authors response: the following sentences was added to the revised manuscript

In the tested period (2003-2007), precipitation reached 1036±653.9 mm/yr, while evaporation losses were 715.9±169.4 mm/yr.

Line 330 – As the water is stored in dams, has the study accounted for evapotranspiration losses?

Authors response: No. The hydrologic model did not go into that detail. It considered evaporation losses at the catchment scale, but not potential amplified evaporation losses from the reservoir following dam construction. It is expected, however, that these losses are low compared to the amplified surface water supply provided by the artificial lake. To clarify this issue we added the following sentence to the revised manuscript, in the description of results following presentation of (new) Figure 8.

These values do not account for potential evaporation losses directly from the reservoir, following dam construction.” We also changed the next sentence, which is now written as

“Regardless the catchment the storage of surface water or groundwater would fulfill present day water deficits and future demands, even considering the aforementioned evaporation losses.”

Line 337 – Please provide reasons as to why the other dams may not be as suitable as the chosen option.

Authors response: To comply with the reviewer comment we added the following sentence to the revised manuscript.

“The use of other catchments is questionable given their low positions in one or more ranks. For example, the Ribeiro do Carvalhal and Ribeiro do Boco catchments are highly positioned in Rank 1 – water availability (positions 8 and 9, respectively), but the negative altitude differences reduces their positions in Rank 3 (positions 1 and 2, respectively) virtually impeding their selection for dam installation considering the potentially high water transport costs. Other catchments are also affected by negative altitude differences, such as the Ribeiro da Peliteira and Ribeiro de Revel (positions 4 and 3 in Rank 3). In these cases, dam installation suitability is further hampered by the low positions in Rank 1 (5 and 2). Rank 2 may hinder the use of Rio Pinhão and Ribeiro dos Rebujais catchments because of their limited forest cover (» 50%; positions 2 and 1). A similar rationale holds for Rio Tinhela catchment (64.6% of forest cover, position 5 in Rank 2). Finally, the Ribeiro do Torno catchment is positioned high in ranks 2 and 3 but low in Rank 1.

Line 352 – Should it be the nitrates loads are higher in surface water (4.7 kg.N/ha) than in groundwater (0.4 Kg.N/ha)?

Authors response: The correction was made.

Reviewer 2 Report

All in all, this seems like a good paper, coherent, well structured, and well argued. Section 2 (conceptual section) could benefit from reflections in relations to previous studies adopting similar approaches such as:

https://link.springer.com/article/10.1007/s12517-018-3433-6

https://link.springer.com/article/10.1007/s12665-018-8031-0

how does this paper relate to these two articles? builds on them? differs?

Once this reflection is added, I think this paper would be of a good standard. 

Author Response

Reviewer #2

All in all, this seems like a good paper, coherent, well structured, and well argued. Section 2 (conceptual section) could benefit from reflections in relations to previous studies adopting similar approaches such as:

https://link.springer.com/article/10.1007/s12517-018-3433-6

https://link.springer.com/article/10.1007/s12665-018-8031-0

how does this paper relate to these two articles? builds on them? differs?

Once this reflection is added, I think this paper would be of a good standard

Authors response: The SWAT model differs substantially from the aforementioned models, because the SWAT is a processed-based semi-distributed hydrologic model while the SPI or the groundwater model described in the above references are not. However, we added the following sentence to the revised manuscript to account for the aforementioned models as potential alternative ways for water resource evaluation.

“The availability of surface water in catchments and groundwater in underlying aquifers can be accessed from a diversity of models [40–46].

Reviewer 3 Report

Keywords: May be “forest” and “water shortage” or “water stress” could be added.

Global comments on the paper:

The paper presents a watershed that suffers from water stress, and evaluates potential new water supply scheme based from rainwater harvesting (more especially from freshwater storage, and also from underground water use, even if this is not particularly defined) . The study is rather clear, and has a robust set of references. The paper is valuable. However, the methodology used for the model and the assessment of the results and discussion suffer from several weaknesses. Then the paper must be improved. Please refer to my hereafter details as main corresponding issues to solve:

Global comment: English should be revised. Wrong wording such as lines 33, 47, 48, 57, 88, 146, 147, 177, 199, 204, 210, 214, 324.

Line 138-141: You should comment that issue. Most of the lost water is from irrigation. It should be replaced by rainwaters or treated wastewaters, and it should be optimized.

Line 153-162: it doesn't mean that there is a direct linkage. You should rather consider the ratio of consumption per capita in these regions.

Line 184: you should provide the volume of transported water per sub region.

Line 185: what is the dam storage capacity?

Line 187: you should present the current water governance. Practically it is not so difficult to treat and distribute water from surface water resources. It is an organizational issue.

Line 195: what is the hypothesis of this scenario? Do you continue the current trend?

Line 198: what is the reason of this 15km2 watershed size?

Line 200-202: This is not clear. A “deficitary” villages is not necessarily directly linked with water supply tanks. You are mixing issues that are different. The transport and storage are more an economic issue, not a problem of water resource insufficiency.

Line 208: SWAT is not an algorithm. It is a model.

Line 208, 220, 221: I suggest you replace climatological by meteorological, climatic by meterological, and so on.

Line 210: it is not « drainage network » but « hydrological network »

Line 230; what about underground aquifer? This is an essential issue if you have forgotten to include underground resources.

Line 234: what do you mean by water tank? Is it dam, concrete tank, drinking water tank, or something else? This is an essential issue, as you seem to consider that a village cannot be supplied without what you call water tank.

Line 250: what is the frequency of data collection? What about solar radiation?

Line 251: Why didn’t you use more recent data than 2007?

Line 252: Regarding the meteorological reference station, could you precise where is it located? Is it far from the studied regions?

Line 272: is it per sub watershed? Please be so kind to add the indication.

Line 272: here appear a reference to underground waters? Could you be more explicit on how you evaluate that.

Line 273: I am not sure to understand your meaning. Practically, the storage is not necessarily possible. You cannot consider the full water fluxes as potentially available. I expect it is not how you have proceeded. Please could you clarify.

Line 278: It is the first time in your paper that you refer to nitrate load. Could you clarify how you have evaluated these values.

Line 284-288: It seems that yow have not imagine that the water from one watershed could be useful for another downstream watershed.

Line 284-288: you should use forest occupancy as surface areas with specific colors rather than circle that could be confused with elevation circles.

Line 284-288: How could it be possible that an outlet of a watershed is at a higher elevation than the village inside this watershed?!

Line 295-297: you should prefer a rank value for each watershed instead of an incremental rank that doesn't mean that the watershed is really interesting.

Line 315-316: your hypothesis seems simple, very simple. You should look at alternative options to evaluate the potential impact on the water stress.

Lines 320-324: the option is still relevant. But it necessitates a modification of the governance. You should argue on that.

Line 328-329: This hypothesis is wrong. Practically if forest can protect water quality, it is clearly not possible to always conclude that it is potable water without dedicated water treatment.

Line 329-330; Here again, the creation of small dams can impact the water quality. It is impossible to conclude of the potability of the raw water. The reverse is probable.

Line 338: is it « of groundwater » or « to groundwater » that has a different meaning ?

Line 340: This is stunning. As you have other watershed with water deficit, it would be much more useful to use that extra available water for other villages rather than developing agricultural activities.

Line 346: How the creation of such a low elevation dam could contribute to local aquifer recharge?! Please clarify.

Line 350-351: Unfortunately, I don’t agree with you. Water quality is not only dependent of Nitrates load. It is inconsistent, more especially in surface freshwaters.

Line 352-353: You are wrong. It is the reverse. I imagine it is a mistake. Please be so kind to amend your text.

Line 358-359: How could you do that if you develop agriculture as you proposed to?? It is rather inconsistent.

Global comment: I suggest you also refer to potential reuse of water that is never referred to in your paper. For instance, you could read this opened access paper (LAFFORGUE M., LENOUVEL V., 2015. Closing the urban water loop: Lessons from Singapore and Windhoek cities. Environmental Science: Water Research and Technology, 2015-1, pp 622-631) and you’ll see that water reuse can reduce water demand from classical water resources.

Author Response

Keywords:

May be “forest” and “water shortage” or “water stress” could be added.

Authors response: we added the suggested keywords to the revised version

Global comments on the paper:

The paper presents a watershed that suffers from water stress, and evaluates potential new water supply scheme based from rainwater harvesting (more especially from freshwater storage, and also from underground water use, even if this is not particularly defined) . The study is rather clear, and has a robust set of references. The paper is valuable. However, the methodology used for the model and the assessment of the results and discussion suffer from several weaknesses. Then the paper must be improved. Please refer to my hereafter details as main corresponding issues to solve:

Authors response: we very much appreciate the effort put in the revision of our manuscript. All comments were carefully attended and incorporated in the revised version.

Global comment: English should be revised. Wrong wording such as lines 33, 47, 48, 57, 88, 146, 147, 177, 199, 204, 210, 214, 324.

Authors response: We revised the entire manuscript to correct grammar or spelling errors, and generally improve the language. We paid especially attention to the aforementioned lines. The changes in the revised manuscript are highlighted in yellow.

Line 138-141: You should comment that issue. Most of the lost water is from irrigation. It should be replaced by rainwaters or treated wastewaters, and it should be optimized.

Authors response: we thank the reviewer comment. We added the following sentences to the revised version.

In the context of conjunctive water management the groundwater currently used for garden irrigation should be replaced by harvested rainwater or treated wastewater [32–36].

Line 153-162: it doesn't mean that there is a direct linkage. You should rather consider the ratio of consumption per capita in these regions.

Authors response: well, we were just trying to explain that people in the municipality are concentrated where groundwater availability is not high and that are mountain regions where water availability is high but population is low. We agree that there is no cause-effect relationship between water availability and population density. It is just a local fact. To clarify the explanation we modified the sentence, which is now written as:

A thorough inspection of this figure reveals a striking reality. The water consumption in villages from the Jales region (Vreia de Jales and Alfarela de Jales) are included in the lowest category (< 30 000 m3/yr), while this region is located where springs can naturally discharge over 100 000 m3/yr of shallow groundwater. In opposition, consumption in Vila Pouca de Aguiar (» 257 000 m3/yr), Bornes de Aguiar / Pedras Salgadas (» 200 000 m3/yr), Capeludos (» 89 000 m3/yr) and Telões (» 85 000 m3/yr), are included in the highest categories, while these villages are located where average spring discharge is low (5 000 – 20 000 m3/yr).

Line 184: you should provide the volume of transported water per sub region.

Authors response: the translocated water was mostly sourced from the Alvão region. We clarified this issue as follows:

In general, these deficits have been satisfied by the municipal authorities through translocation of water from the Alvão region using vehicle water tanks.

Line 185: what is the dam storage capacity?

Authors response: we clarified this point adding the sentence

“(e.g., the Pinhão River dam with a storage capacity of 4.2 Mm3; http://www.somagueNaN/portfolio_detail/barragem-do-pinhao/)

Line 187: you should present the current water governance. Practically it is not so difficult to treat and distribute water from surface water resources. It is an organizational issue.

Authors response: we clarified the issue by completing the sentence, which is now written

“Besides the quality issue, the large dam solution would eventually imply a loss of municipal autonomy concerning drinking water supply to the population, because large dams are frequently explored by private companies

Line 195: what is the hypothesis of this scenario? Do you continue the current trend?

Authors response: Yes, the hypothesis is based on the past precipitation trend (Figure 2) and consumption trend (Figure 5)

Line 198: what is the reason of this 15km2 watershed size?

Authors response: The purpose was to restrict the modeling to small catchments that would be able to ensure a “single” use, namely forest use, because forest cover was considered a key suitability parameter. Larger watersheds are less likely to accomplish that. However, that was not directly imposed to the modeling and therefore is not strictly a area threshold. Therefore, we removed the sentence from the revised manuscript. The modeled catchments were indeed all < 15 km2 but that occurred naturally, not imposed as a priori condition.

Line 200-202: This is not clear. A “deficitary” villages is not necessarily directly linked with water supply tanks. You are mixing issues that are different. The transport and storage are more an economic issue, not a problem of water resource insufficiency.

Authors response: we agree that the text was not clear. The whole paragraph was re-written for clarification. Now the text is:

In this study, the proposed alternative source was stream water stored in small dams installed in small catchments and explored by the municipal authorities. The selection of sites was based on three suitability parameters: water availability, dominant land cover and altitude difference between the catchment outlet and drinking water tanks used to supply the towns of Vila Pouca de Aguiar and Bornes de Aguiar / Pedras Salgadas. The suited sites will be located where high rainfall ensures large stream flows, forest land cover makes certain good quality water, and the altitude of catchment outlets allows the transport of water by gravity to the application areas reducing costs.

Line 208: SWAT is not an algorithm. It is a model.

Authors response: We replaced “algorithm” by “model”, as suggested.

Line 208, 220, 221: I suggest you replace climatological by meteorological, climatic by meterological, and so on.

Authors response: Done

Line 210: it is not « drainage network » but « hydrological network »

Authors response: Replaced

Line 230; what about underground aquifer? This is an essential issue if you have forgotten to include underground resources.

Authors response: The modeling exercise considered the underground resources. They appear in Table 2 under the heading “Water availability”.

Line 234: what do you mean by water tank? Is it dam, concrete tank, drinking water tank, or something else? This is an essential issue, as you seem to consider that a village cannot be supplied without what you call water tank.

Authors response: The term water tank refers to drinking water tank (we clarified the term in the revised version). The drinking water tank issue is not related to water supply capacity of these tanks but with the altitude at which they are installed, compared to the altitude of the catchment outlets. If the purpose is to transport water from the catchments to the water tanks it is important that the later are installed at lower altitude than the former so the transport is done by gravity reducing the costs.

Line 250: what is the frequency of data collection? What about solar radiation?

Authors response: we clarified the issue as follows:

“The meteorological data (daily precipitation and long-term average temperature, humidity, wind speed and solar radiation) were downloaded from the National Water Resources Information System (http://snirhNaN/), and span the 2003-2007 period.”

Line 251: Why didn’t you use more recent data than 2007?

Authors response: The meteorological data spans the period for which we could compile data on water consumption.

Line 252: Regarding the meteorological reference station, could you precise where is it located? Is it far from the studied regions?

Authors response: The meteorological stations are located within the municipality limits, as illustrated in Figure 9a.

Line 272: is it per sub watershed? Please be so kind to add the indication.

Authors response: We clarified the issue by changing the sentence:

“In the modeled catchments, the maximum available surface water can reach 5 Mm3/yr (Ribeiro do Boco), while groundwater will not exceed 3Mm3/yr (Ribeiro do Boco and Ribeiro do Cabouço).”

Line 272: here appear a reference to underground waters? Could you be more explicit on how you evaluate that.

Authors response: ArcSWAT estimates runoff as well as groundwater flow (see figure 7). That is why we present results for runoff and groundwater in Table 2.

Line 273: I am not sure to understand your meaning. Practically, the storage is not necessarily possible. You cannot consider the full water fluxes as potentially available. I expect it is not how you have proceeded. Please could you clarify.

Authors response: we agree. We clarify that the obtained volumes are maximum storable values, in absence of any type of loss. To do that we added the following sentence

“These values do not account for potential evaporation losses directly from the reservoir, following dam construction. They also neglect losses related with ecological flows.

Besides, we completed the following sentence

“Regardless the catchment, the storage of surface water or groundwater would fulfill present day water deficits and future demands, even considering the aforementioned losses.”

Line 278: It is the first time in your paper that you refer to nitrate load. Could you clarify how you have evaluated these values.

Authors response:  That issue has been clarified in the original version (discussion), in the sentence “Water quality has not been measured in the Cabouço catchment in this study. But ArcSWAT provided estimates for annual nitrate loadings (Kg N/ha) in surface water and groundwater (Figure 9), based on catchment land cover.”

Line 284-288: It seems that yow have not imagine that the water from one watershed could be useful for another downstream watershed.

Authors response:  The target areas as regards water transfer are the towns of Vila Pouca de Aguiar and Bornes de Aguiar / Pedras Salgadas, located in the lowlands. The purpose was not a generalized transfer of water across watersheds but the identification of headwater catchments that could deliver clean water (drinking water) to the aforementioned towns

Line 284-288: you should use forest occupancy as surface areas with specific colors rather than circle that could be confused with elevation circles.

Authors response:  Figure 10 represents a lot of related things. The representation of forest percentage with graduated grey symbols was the best compromise to include all themes in the figure. This time we could not attend the reviewer suggestion.  

Line 284-288: How could it be possible that an outlet of a watershed is at a higher elevation than the village inside this watershed?!

Authors response:  Please note that the target villages are represented by their drinking water tanks (white circles in Figure 10), which are located outside the potential source catchments. The altitude differences between catchment outlets (green circles) and drinking water tanks (white circles) are represented by the colored circles and can be positive or negative.

Line 295-297: you should prefer a rank value for each watershed instead of an incremental rank that doesn't mean that the watershed is really interesting.

Authors response:  In the revised version we still used the incremental rank but analyzed the individual ranks separately to conclude about the basins’s suitability. The clarification is presented in the following sentence:

“According to this combined assessment, the most suitable catchment to install a small dam aiming the supplement of drinking water to the villages of Vila Pouca de Aguiar and Bornes de Aguiar / Pedras Salgadas is the Cabouço catchment. This basin was ranked 7 on surface water availability (2.4 Mm3/yr), 8 on forest cover (84.8%) and 6 in altitude difference (86.8 m), and would be a nice place to store quality water in the Vila Pouca Municipality. The use of other catchments is questionable given their low positions in one or more ranks. For example, the Ribeiro do Carvalhal and Ribeiro do Boco catchments are highly positioned in Rank 1 – water availability (positions 8 and 9, respectively), but the negative altitude differences reduces their positions in Rank 3 (positions 1 and 2, respectively) virtually impeding their selection for dam installation considering the potentially high water transport costs. Other catchments are also affected by negative altitude differences, such as the Ribeiro da Peliteira and Ribeiro de Revel (positions 4 and 3 in Rank 3). In these cases, dam installation suitability is further hampered by the low positions in Rank 1 (5 and 2). Rank 2 may hinder the use of Rio Pinhão and Ribeiro dos Rebujais catchments because of their limited forest cover (» 50%; positions 2 and 1). A similar rationale holds for Rio Tinhela catchment (64.6% of forest cover, position 5 in Rank 2). Finally, the Ribeiro do Torno catchment is positioned high in ranks 2 and 3 but low in Rank 1.

Line 315-316: your hypothesis seems simple, very simple. You should look at alternative options to evaluate the potential impact on the water stress.

Authors response: We are not sure about the meaning the reviewer gave the “water stress”. Is it about water stress in soil? The focus of this paper is on drinking water supply, solely.

Lines 320-324: the option is still relevant. But it necessitates a modification of the governance. You should argue on that.

Authors response:  We fully agree. We added the following sentence and reference about that.

“As a matter of fact, the option is relevant but requires a modification of governance [21]. To our view, the option is relevant but requires modification of governance in directions comparable to other cases [56–59].

Line 328-329: This hypothesis is wrong. Practically if forest can protect water quality, it is clearly not possible to always conclude that it is potable water without dedicated water treatment.

Authors response:  we changed “potable” by “good”

Line 329-330; Here again, the creation of small dams can impact the water quality. It is impossible to conclude of the potability of the raw water. The reverse is probable.

Authors response:  we changed “potable” by “good quality”

Line 338: is it « of groundwater » or « to groundwater » that has a different meaning ?

Authors response:  it is of groundwater, because the catchment outlet discharges stream water composed of surface and groundwater components.

Line 340: This is stunning. As you have other watershed with water deficit, it would be much more useful to use that extra available water for other villages rather than developing agricultural activities.

Authors response:  well, the actual and future deficits in Vila Pouca de Aguiar are < 0,1Mm3/yr, while the small Cabouço catchment would be able to store 3Mm3/yr of good quality water. We believe this is enough reason to consider the multiple use of this water.

Line 346: How the creation of such a low elevation dam could contribute to local aquifer recharge?! Please clarify.

Authors response:  The catchment is located at 900 m of altitude, which is a high altitude in the region. The catchment belongs to the Alvão mountains and is a headwater catchment. We have little doubt that the creation of a dam here would improve recharge.

Line 350-351: Unfortunately, I don’t agree with you. Water quality is not only dependent of Nitrates load. It is inconsistent, more especially in surface freshwaters.

Authors response: We accept. Nitrate is an indicator of water quality, among other indicators. Since ArcSWAT estimates nitrate loads based on land cover, we presented those results to indicate that, probably, water quality in the Cabouço catchment is good. We do not have any other information on water quality in this catchment. Therefore, we rephrased the sentence:

“ArcSWAT provided estimates for annual nitrate loadings (Kg N/ha) in surface water and groundwater (Figure 9) based on catchment land cover, which can be used as indicators of water quality. The estimated loads are higher in groundwater (0.4 Kg.N/ha) than in surface water (4.7 Kg.N/ha), but are generally low.”

Line 352-353: You are wrong. It is the reverse. I imagine it is a mistake. Please be so kind to amend your text.

Authors response: Thanks for the attention. We corrected (reversed) the values.

Line 358-359: How could you do that if you develop agriculture as you proposed to?? It is rather inconsistent.

Authors response:  Agriculture is to be developed in the South valley identified in Figure 10 by the green dashed area, not in the Cabouço catchment.

Global comment: I suggest you also refer to potential reuse of water that is never referred to in your paper. For instance, you could read this opened access paper (LAFFORGUE M., LENOUVEL V., 2015. Closing the urban water loop: Lessons from Singapore and Windhoek cities. Environmental Science: Water Research and Technology, 2015-1, pp 622-631) and you’ll see that water reuse can reduce water demand from classical water resources.

Authors response:  The reference was added as suggested.

Round 2

Reviewer 1 Report

The manuscript is much improved following authors' taking reviewers' comments into considerations. 

Please complete a final English check.

Author Response

We have revisited the entire manuscript to detect any spelling errors or grammar tenses, and prepared a final manuscript accordingly. Many thanks again for the efforts made and time dedicated to the review of this manuscript.

Reviewer 3 Report

Thanks to the authors for the revised version that includes some improvements. However, there are still some modifications to perform, more especially on requests that have not been considered in the revised paper. Please refer to my hereafter details as main corresponding issues to solve:

Global comment: English should still be improved.

Line 203: You should indicate the hypothesis of these scenarios “5-15” in your paper.

Line 230-263; Regarding underground aquifer, I think you have misunderstood my comment. It seems that you have no real data on the existing aquifers, then that your model is more theoretical than effectively considering real existing aquifers. Maybe you could at least comment that in your discussion part as a limitation of your work.

Line 273-282: Regarding the meteorological reference stations, Your answer refer to figure 9a but this figure doesn’t include the reference stations.  Maybe you could indicate in your paper that the reference meteorological stations are inside the global considered catchment.

Line 307: It is the first place in your paper that you refer to nitrate load. In your methodological part 2, you should add some indications on how you have evaluated these loads (because it is only presented lines 386-392 in the discussion part).

Line 350-351: Your hypothesis seems simple. Please could you at least indicate the origin of these trends. I understand that you have not looked at alternative trends to evaluate the potential impact of these 2 values on the water stress.

Line 377: My comment on agriculture use is still pertinent (refer to your lines 396-397 that are not necessarily in agreement with a strong agriculture development). Indeed, even if you consider that the contamination will be downstream of Cabouço catchment, my concern is rather philosophical, as it could lead to strong water issue downstream of your watershed. To avoid that, I suggest you suppress “, namely agriculture” from your sentence.

Line 383-384: Regarding the creation of such a low elevation dam contributing to local aquifer recharge. Practically, it is not automatic. It depends of the aquifers and of the soil properties. But if you are sure that in your case it sustains aquifer recharge, just keep it.

Line 387-392: You know my point of view on your simplified approach of water quality. Your approach is very simple and is not strong enough to conclude on water quality. I suggest you temper your conclusion on nitrates by indicating that it is only a rough indication, and that a specific dedicated study should be performed for a better assessment.  

Reference 36- line 511: replace “2015” by “2015-1, 622-631”.

Author Response

We again, thank all the effort and careful attention over our manuscript. This reviewer made a very nice job and we tried to answer positively to all his/her comments.

Global comment:

English should still be improved.

Authors’ response

The text was again revisited and improved where grammar tenses or spelling errors were detected

Comment #1

Line 203: You should indicate the hypothesis of these scenarios “5-15” in your paper.

Authors’ response

The hypothesis is now indicated in the Table legend, as follows: The scenario “5-15” refers to the expected 5% consumption increase and 15% precipitation decrease over the next two decades.

Comment #2

Line 230-263; Regarding underground aquifer, I think you have misunderstood my comment. It seems that you have no real data on the existing aquifers, then that your model is more theoretical than effectively considering real existing aquifers. Maybe you could at least comment that in your discussion part as a limitation of your work.

Authors’ response

May be the reviewer is talking about data on hydraulic heads or aquifer hydraulic conductivity and effective porosity. These data are required to run flow models such as MODFLOW but not to run ArcSWAT since the groundwater component is estimated by difference using the water balance equation. We do have values for aquifer formation constants, but we did not used in the SWAT model because they are not necessary.

Comment #3

Line 273-282: Regarding the meteorological reference stations, Your answer refer to figure 9a but this figure doesn’t include the reference stations.  Maybe you could indicate in your paper that the reference meteorological stations are inside the global considered catchment.

Authors’ response

In fact, our answer refers to Figure 8a (not Figure 9a) which do include the reference meteorological stations.

Comment #4

Line 307: It is the first place in your paper that you refer to nitrate load. In your methodological part 2, you should add some indications on how you have evaluated these loads (because it is only presented lines 386-392 in the discussion part).

Authors’ response

We added the following sentence to the methodological part “Based on the indicated land uses, nutrient loads (e.g., nitrate) are also estimated using solute transport models.

We also adjusted Figure 7 to the new text.

Comment #5

Line 350-351: Your hypothesis seems simple. Please could you at least indicate the origin of these trends. I understand that you have not looked at alternative trends to evaluate the potential impact of these 2 values on the water stress.

Authors’ response

The hypothesis is based on the past trends represented in Figures 2 and 5a that are anticipated for the near future. This is our best guess about rainfall evolution and water consumption. We clarified the sentence in the revised version, which is now written as:

The actual deficits are greater than 55 000 m3/yr and can increase to approximately 90 000 m3/yr in the near future (next two decades) if scenarios of rainfall decrease (–7.5%/decade) and consumption increase (+2.5%/decade) become reality, as can be anticipated from past trends (Figure 2 and 5a, respectively).”

Comment #6

Line 377: My comment on agriculture use is still pertinent (refer to your lines 396-397 that are not necessarily in agreement with a strong agriculture development). Indeed, even if you consider that the contamination will be downstream of Cabouço catchment, my concern is rather philosophical, as it could lead to strong water issue downstream of your watershed. To avoid that, I suggest you suppress “, namely agriculture” from your sentence.

Authors’ response

The term “, namely agriculture” was suppressed from the revised version.

Comment #7

Line 383-384: Regarding the creation of such a low elevation dam contributing to local aquifer recharge. Practically, it is not automatic. It depends of the aquifers and of the soil properties. But if you are sure that in your case it sustains aquifer recharge, just keep it.

Authors’ response

We keep it

Comment #8

Line 387-392: You know my point of view on your simplified approach of water quality. Your approach is very simple and is not strong enough to conclude on water quality. I suggest you temper your conclusion on nitrates by indicating that it is only a rough indication, and that a specific dedicated study should be performed for a better assessment. 

Authors’ response

The following sentence was added to the revised version

It is worth to note that the estimated nitrate loads are rough indications of real loads because they could not be calibrated with measured data, and therefore a specific dedicated study should be performed for a better assessment.

Comment #9

Reference 36- line 511: replace “2015” by “2015-1, 622-631”.

Authors’ response

The reference was corrected